# Monocular 3D Human Pose Estimation via Euler Angles

## Abstract

Monocular 3D human pose estimation is a key challenge in computer vision. While existing joint position-based methods often struggle with accurate bone length prediction and rotation ambiguities under joint collinearity, joint rotation-based methods circumvent the bone length issue but face discontinuity problems when regressing the body's self-rotation angle, limiting their practical application. In this work, we theoretically analyze the root cause of this discontinuity and propose a framework for Euler angle-based pose estimation. Our method transforms Euler angle into its corresponding sine-cosine representation on the unit circle, ensuring a continuous and wrap-around-free formulation, and introduces body orientation as an internally learned conditional variable, which jointly optimizes the prediction of Euler angles in a unified learning process. By allowing the network to adapt its regression behavior based on the predicted condition, the framework effectively resolves the discontinuity inherent in Euler-angle regression. Experiments across diverse model architectures, including CNNs, GCNs, and Transformers, demonstrate that our approach enables continuous and stable self-rotation prediction. The framework forms a versatile and efficient module that maintains compatibility with existing backbones and facilitates streamlined end-to-end training.

## 1 Introduction

Monocular 3D human pose estimation is a prominent research direction in computer vision with wide applications in virtual reality, augmented reality, avatar generation, and metaverse technologies(Yang et al., 2022; Anvari et al., 2023; Lu et al., 2024; Li et al., 2025). By leveraging RGB or RGB-D images and video sequences, monocular 3D human pose estimation can reconstruct the skeletal joint positions and motion in 3D space, which serves as a crucial bridge between the physical and digital worlds.

Current research can be broadly categorized into joint position-based and joint rotation-based methods. Joint position-based methods estimate the 3D coordinates of human joints in the world coordinate system to reconstruct the skeleton and motion (Zheng et al., 2023; 2021; Cai et al., 2019; Cheng et al., 2020). However, joint position-based methods face two main challenges: (1) the skeletal bone lengths often appear inconsistent, making it difficult to accurately recover joint locations. To alleviate this issue, several studies constrain bone lengths within a plausible range to improve accuracy (Chen et al., 2022; Kang et al., 2023). (2) These methods also struggle to infer self-rotation when joints are collinear. Fisch & Clark (2021) proposed virtual markers to approximate rotations along the bone axes, however, their approach remains limited by image resolution and the lack of texture information.

Joint rotation-based methods can avoid the bone length inconsistency issue (Jiang et al., 2022). However, they suffer from discontinuities caused in rotation angles by wrap-around (Pavllo et al., 2018; Zhou et al., 2019; Pepe et al., 2024). For example, after a full rotation, the angle abruptly resets to zero and leads to discontinuous predictions of self-rotation. Such wrap-around signals are difficult for neural networks to model, restricting the applicability of rotation-based methods. Common rotation representations include Euler angles (Diebel, 2006), quaternions (Pavllo et al., 2018), 6D rotation (Zhou et al., 2019), and axis-angle (Loper et al., 2015) formats. Among them, Euler angle is more intuitive and parameter-efficient compared to other forms, making it widely

adopted in character animation and motion editing. Motivated by this inherent interpretability, we revisit human pose estimation from the perspective of Euler-angle modeling.

In this paper, we focus on Euler angle-based pose estimation and analyze the underlying cause of discontinuities in self-rotation prediction. While the body's self-rotation is continuous in 3D space, its Euler angle introduces discontinuous jumps. To address this, we propose a framework for Euler-angle estimation. We interpret the discontinuity as a wrap-around effect arising from mapping the Euler-angle representation on the SO(3) manifold into its sine-cosine form on an SO(6) manifold, and introduce body orientation as an internally learned conditional variable to facilitate a more coherent and continuous manifold mapping. The model jointly predicts the angles and the orientation condition, enabling stable and continuous self-rotation estimation. Our framework is lightweight and can be integrated into existing 3D pose estimation backbones, including Pavllo et al. (2019); Zheng et al. (2021); Zhao et al. (2023; 2019); Liu et al. (2020b;a).

Our contributions are summarized as follows:

- We provide a theoretical analysis of the wrap-around problem in Euler angle representations, modeling it via a 3D helix. Through spatial projection, we transform continuous 3D rotation into a learnable sine-cosine form representation suitable for neural networks.

- We propose a conditional Euler-angle learning framework that treats body orientation as an internally learned variable. The model predicts this condition and Euler angles jointly, enabling end-to-end training and continuous angle regression.

- We demonstrate efficient transfer from position-based models to rotation-based ones. Our method allows the current models to transfer from joint position to joint rotation representations, enhancing the completeness of 3D pose estimation.

The remainder of this paper is organized as follows: Section 2 reviews related works on joint position-based and joint rotation-based methods. Section 3 presents the theoretical foundation of our approach and introduces a framework that jointly predicts Euler angles and internal orientation conditions to achieve continuous rotation poses. Section 4 presents the experimental setup and results, followed by a discussion evaluating the proposed method across a range of pose estimation models.

## 2 RELATED WORK

**Joint position-based estimation.** (1) CNN-based methods. Pavllo et al. (2019) extended the temporal receptive field through multi-layer dilated convolutions with residual connections, thereby enhancing temporal correlations in pose sequences. Chen et al. (2021) further decoupled the task into bone length and joint direction learning to improve consistency in bone length prediction. Chen et al. (2022) leveraged bone length invariance constraints to refine 3D pose regression. To resolve rotation ambiguities in joint position representations, Fisch & Clark (2021) introduced virtual markers to model joint roll rotations. (2) GCN-based methods. Zhao et al. (2019) incorporated prior semantic information among joints to improve spatial feature modeling. Yu et al. (2023) captured global correlations via adaptive graph convolutions and refined local features through independent connection layers. Ci et al. (2019) designed a Local Connection Network (LCN) to strengthen the modeling of local spatial dependencies. (3) Transformer-based methods. Zheng et al. (2021) demonstrate the effectiveness of a pure Transformer architecture for 3D human pose estimation. Li et al. (2022) learned spatio-temporal information via multi-hypothesis generation and feature fusion. Zhang et al. (2022) achieved multi-level spatio-temporal separation and fusion by alternately stacking spatial and temporal Transformer blocks. Kang et al. (2023) proposed a dual-chain design (local-to-global and global-to-local) to fully capture complex multi-level dependencies among human joints. Shuai et al. (2023) adaptively fused multi-view and temporal features to handle varying views and video lengths without camera calibration. In summary, research on joint position-based estimation primarily focuses on temporal modeling, spatial structure constraints, and capturing both global and local dependencies. Different networks architectures (CNNs, GCNs, Transformers) have continuously improved the representation of spatio-temporal dependencies among joints. Some works attempt to alleviate issues, such as inconsistent bone lengths and missing self-rotation, but these challenges remain largely unsolved.

**Joint rotation-based estimation**. Akhter & Black (2015) investigated the rotation limits of human joints under specific poses and introduced limit-value constraints. Building on this, Yang et al. (2023) incorporated human pose priors via learnable rotation tokens to constrain estimated angles within plausible ranges. To mitigate error accumulation along kinematic chains, Pavllo et al. (2018) employed recurrent neural networks to estimate joint rotations and introduced a differentiable forward kinematics loss to mitigate accumulated errors. However, joint rotation representations suffer from discontinuities. Several strategies have been proposed to solve the problem. Burgermeister & Curio (2022) adopted spherical coordinates, with the polar angle represents pitch and the azimuth angle represents horizontal orientation. Li et al. (2021) decomposed rotations into twist and swing components, estimated respectively via numerical computation and neural prediction. Zhou et al. (2019) proposed the 6D rotation representation and Banik et al. (2024) further utilized 2D rotation information from 2D keypoints to assist 3D rotation estimation. In summary, discontinuities in rotation-based representations are typically addressed in two ways: (1) Reducing rotational degrees of freedom using adopting 2D angle representations, which avoids discontinuities but sacrifices one degree of freedom, limiting the ability to represent complex rotations; (2) Employing continuous representations, which preserve full rotational information but increase parameter complexity and may introduce errors.

## 3 METHOD

### 3.1 THESIS ANALYSIS

To regress the sequence of human pose, we aim to learn a continuous function for the body rotation:

$$\theta = f_w(x) \tag{1}$$

Here, $\theta$ lies on a helix, with body rotation defined over $[0, +\infty)$. The ground-truth value typically falls within $[\theta, 2n * 180]$, where $n$ denotes the number of full rotations. However, from a single image, $f_w(x)$ can only predict from $0°$ to $360°$, since the total number of full rotations cannot be inferred without prior information.

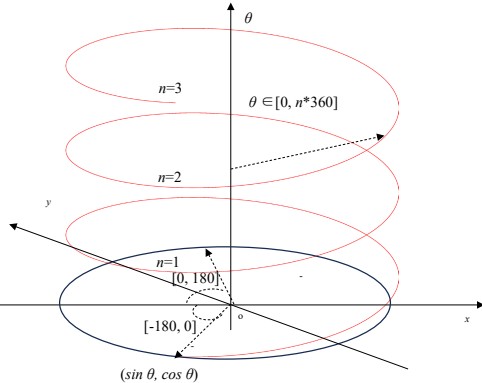

Figure 1: The body rotation of the human pose follows a helical trajectory.

As shown in Figure 1, there is a gap between each round in a helix. The network predicts the continuous rotation value, which can cause wrap-around issues at the gap. By projecting the helix into 2D Cartesian coordinates, the regression interval from $[0, 2n \cdot 180)$ is mapped to $[0, 360)$, and the helix is transformed into a unit circle. The angular $\theta$ is mapped to its sine-cosine embedding on the unit circle. After shifting $180°$, the range can be divided into two intervals: $[-180, 0)$, whose signs along the $y$-axis provide an interpretable indicator of global body orientation. To generalize the formula, Eq. (1) can be rewritten as below:

$$(\sin\theta, \cos\theta) = f_w(x, y) \tag{2}$$

By leveraging the orientation cue as an additional prior, the model is better regularized during training, which enhances its capability to construct stable high-dimensional manifold mappings, effectively transforming the rotation representation from the SO(3) manifold to the SO(6) manifold for 3D Euler angles (Zhou et al., 2019). Consequently, the network can be guided through the $y$-value to learn more consistent and continuous rotational representations.

## 3.2 CONDITIONAL POSE ESTIMATION

The overall framework of our conditional Euler-angle-based pose estimation is illustrated in Figure 2. Our method is designed as an end-to-end model and consists of the following components: (1) Integrated condition learning: Given the input 2D keypoints, the network simultaneously predicts an learnable orientation condition prior while regressing pose parameters. The orientation condtion prior is refined during training and guides the network to produce continuous Euler-angle outputs. (2) Pose estimation with conditional guidance: The learned orientation prior is integrated with the input 2D keypoints and serves as a continuous guiding signal for Euler-angle regression in a fully differentiable manner. The framework produces outputs as 7D that includes sine and cosine values(6D) of Euler angles and the orientation conditions(1D). (3) Architecture compatibility: The unified design are compatible with existing backbone networks for 3D human pose estimation, including CNN-, GCN-, and Transformer-based models, and can be trained end-to-end without requiring a separate classification stage.

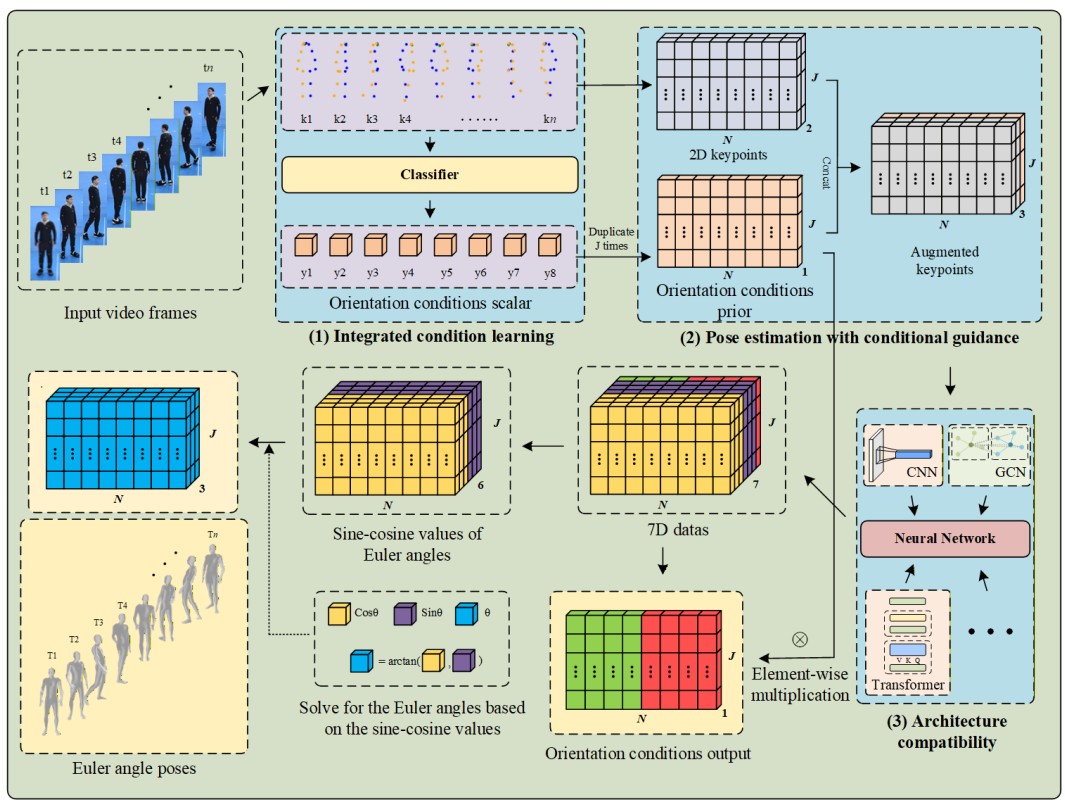

Figure 2: The overall framework of conditional Euler angle-based human pose estimation.

### 3.2.1 CONDITION PRIOR LEARNING

In rotation-based methods, the overall body rotation is determined by the horizontal rotation of the root joint. The horizontal rotation angle can increase continuously and span multiple cycles. Therefore, it is difficult to estimate the exact number of full rotations using only a limited set of frames.

As illustrated in Figure 3(a), we implement a conditional Euler angle learning method by projecting the root joint's angle onto the 2D plane, allowing the angle can be divided into two intervals according to the body's orientation. In Figure 3(b), we present the condition results of different human body orientations. For representing the self-rotation of each pose in the camera view, the projected angle in the world coordinate system is denoted as $\phi \in [-180, 180]$. Based on the interval of $\phi$, a conditional value $y$ is derived in a guided manner from the corresponding discrete orientation value.

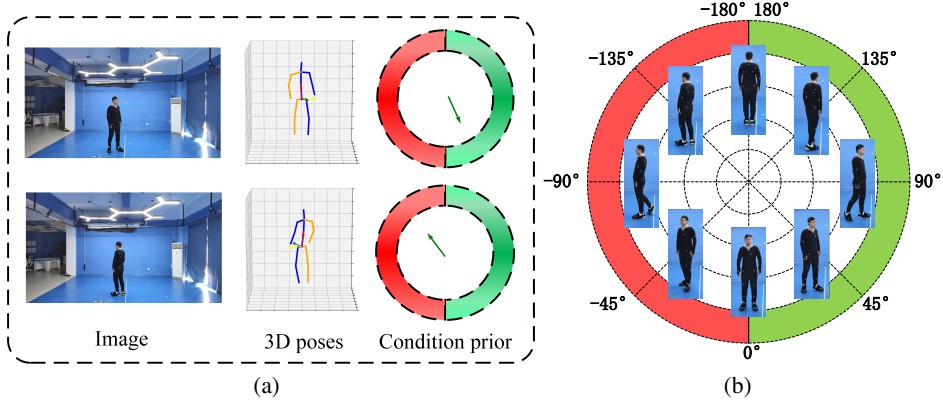

Figure 3: (a) Condition prior learning. The horizontal Euler angle is divided into two intervals based on the body's orientation. (b) Angular intervals corresponding to the conditional Euler angles. The green interval spans [0, 180], while the red interval covers [-180, 0].

To learn the condition information, we employ a lightweight ResNet-inspired network integrated directly into the framework. Our network is adapted to accept pose sequences of shape $B \times N \times J \times 2$. At the output stage, the prediction head is replaced with a compact conditional feature of size $B \times N \times 1$.

As formulated in Eq. (2), the orientation conditional prior is learnable and optimized jointly with the pose parameters. In our framework, for a sequence of $N$ frames, the network inherently learns a latent conditional representation. This condition prior is fused with the 2D keypoint input to provide orientation-aware guidance throughout the pose estimation pathway. In this way, the network is able to regress Euler angles end-to-end, with the conditional feature dynamically shaping the learning of pose parameters. The conditional guidance is learned jointly as part of the overall optimization objective, enabling the network to effectively capture orientation priors.

$$P_{\text{ex}} = \text{concat}(P, c_J) \in \mathbb{R}^{N \times J \times 3} \tag{5}$$

For each joint in each frame, the network learns a learnable orientation prior that captures the body's global orientation. Then, they are concatenated again with the original input 2D keypoints within the network. By incorporating this learnable orientation prior, the model receives continuous guidance on body orientation, which substantially improves its ability to regress Euler angles smoothly and handle rotational discontinuities.

### 3.2.2 SINE-COSINE VALUES OF EULER ANGLE

Instead of predicting Euler angles directly, we predict their sine and cosine values. This design addresses the numerical discontinuity problem inherent in Euler angle regression. For example, if the ground truth is $-179°$ and the predicted value is $170°$, the original MPJASE loss would compute an error of $349°$, whereas the actual angular difference modulo $360°$ is only $11°$.

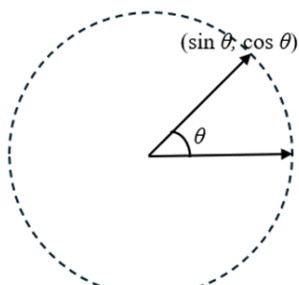

Figure 4: Sine-cosine values of Euler angle.

As shown in 4, each Euler angle $\theta$ is represented by its sine and cosine values $(\sin\theta, \cos\theta)$, and the final angle is recovered via the arctangent function $\theta = \arctan\left(\frac{\sin\theta}{\cos\theta}\right)$. This transformation increases the output dimension from 3 to 6 if each Euler angle is mapped onto a 2D unit circle.

### 3.2.3 LOSS FUNCTION

For rotation-based pose estimation within our framework, we employ a composite loss function to jointly optimize the Euler angle regression and the internal orientation-condition learning.

The primary loss for pose regression is the Mean Per Joint Angular Separation Error (MPJASE), which quantifies the rotational discrepancy between predictions and ground truth, as defined in Eq. (6).

$$\mathcal{L}_{\text{MPJASE}} = \frac{1}{N \times J} \sum_{i=1}^{N} \sum_{j=1}^{J} \|e_{i,j} - \hat{e}_{i,j}\|_1 \tag{6}$$

Equation (6) computes the average $L_1$ distance between the predicted Euler angle vector $\hat{e}i, j \in \mathbb{R}^3$ (comprising $\alpha, \beta, \gamma$) and its ground-truth counterpart $ei, j$ across all $J$ joints and $N$ frames.

To guide the learning of the internal orientation representation—which replaces the external classifier—a Binary Cross-Entropy (BCE) loss is applied to the model's conditional scalar output, aligning it with the discretized orientation intervals. The total training objective is the weighted sum of both components:

$$\mathcal{L}_{\text{Total}} = \mathcal{L}_{\text{MPJASE}} + \lambda \cdot \mathcal{L}_{\text{BCE}} \tag{7}$$

where $\lambda$ is a balancing hyperparameter. This combined loss enables simultaneous and end-to-end optimization of continuous pose estimation and its conditioning prior within a single network.

## 4 EXPERIMENTAL RESULTS AND DISCUSSION

### 4.1 EXPERIMENT SETUP

To validate the effectiveness of the proposed method, we conducted a comprehensive evaluation on our custom dataset, which provides synchronized Euler angle annotations, body orientation information and 2D keypoints (details in Appendix A). The experimental protocol consists of two parts: (1) Integrating the proposed conditional Euler angle representation into multiple mainstream 3D human pose estimation models to examine its generalization ability and its capacity to mitigate rotational discontinuities across diverse architectures. These evaluations collectively demonstrate the robustness and versatility of our approach. (2) Performing an ablation study to analyze different ways of predicting Euler angles and conditional information influence the stability of rotation representation. This includes comparisons between direct angle prediction, classifier-assisted prediction, and our unified conditional framework.

## 5 EXPERIMENTAL VALIDATION ON MULTIPLE BASELINES

We demonstrate our results on a broader set of baseline models, including Pavllo et al. (2019); Zhao et al. (2019); Zheng et al. (2021); Zhao et al. (2023); Liu et al. (2020b;a).Our new method achieves significant improvements, not only outperforming the original approaches overall but also correctly estimating previously challenging ambiguous poses. As illustrated in Figure 5, we compare the performance of different methods on two representative pose types.

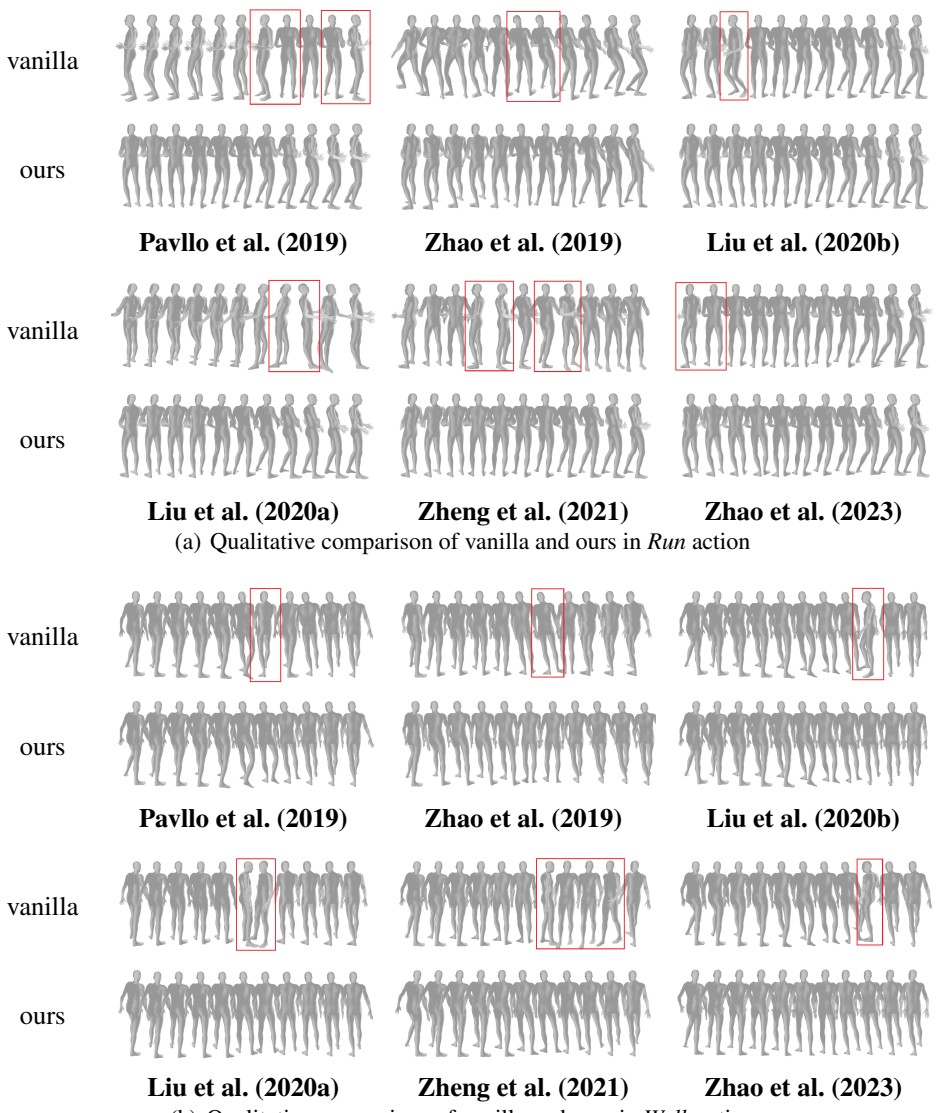

(a) Qualitative comparison of vanilla and ours in *Run* action

(b) Qualitative comparison of vanilla and ours in *Walk* action

Figure 5: Qualitative comparison of vanilla and ours across different methods for *Run* and *Walk* actions.

We further analyze the performance of different methods at rotational discontinuities. Figure 5 shows consecutive frames of *Run* and *Walk* actions during abrupt rotation changes. In the *Run* action (Figure 5(a)), Pavllo et al. (2019) and Liu et al. (2020a) exhibit sharp rotational jumps without smooth transitions, while Zheng et al. (2021) shows over- or under-rotation across several frames. Liu et al. (2020b) and Zhao et al. (2023) generally produce smoother transitions but still have frame-level errors, whereas Zhao et al. (2019) causes unnatural body tilting. In the *Walk* action (Figure 5(b)), Pavllo et al. (2019); Liu et al. (2020b); Zheng et al. (2021); Zhao et al. (2023); Liu et al.

(2020a) display directional errors in some frames, with Zhao et al. (2019) again exhibiting tilting. These observations highlight the limitations of Euler-angle representations, which often yield discontinuous predictions at interval boundaries. By incorporating conditional estimation, both actions achieve smooth and coherent transitions across all six methods, demonstrating the effectiveness and generality of our approach.

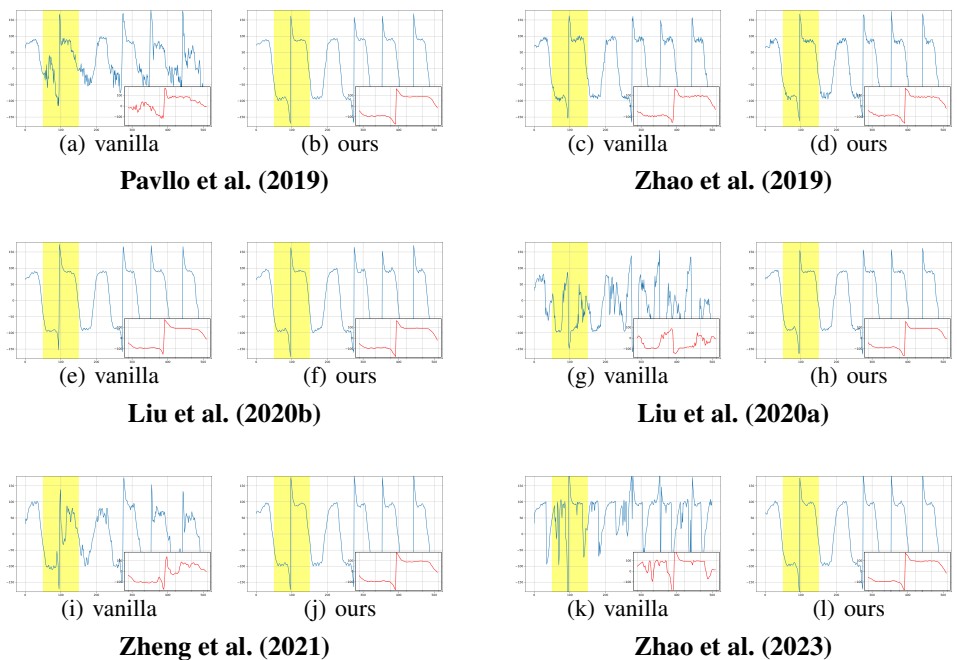

Figure 6: Comparison of horizontal rotation angles. The yellow region indicates the angle range from −180° to 180°. Pavllo et al. (2019); Zheng et al. (2021); Zhao et al. (2023); Liu et al. (2020a) fail to capture the step changes, Zhao et al. (2019); Liu et al. (2020b) exhibit oscillations. With our method, all models produce continuous rotations.

To further validate the effectiveness of our method, Figure 6 illustrates the curves of horizontal rotation angle curves over frames for different approaches. The results reveal a typical periodic wrap-around phenomenon. Without conditional estimation, the predicted curves fail to correctly follow the ground truth at step changes. Specifically, Pavllo et al. (2019); Zheng et al. (2021); Zhao et al. (2023); Liu et al. (2020a) exhibit strong fluctuations around the discontinuities, indicating difficulty in modeling rotational step changes. Zhao et al. (2019); Liu et al. (2020b) produce curves closer to the ground truth, but their step transitions occur a few frames earlier, resulting in inaccurate pose predictions. After applying our method, the predictions accurately reconstruct the step transitions. These results confirm that the proposed conditional method effectively address the challenge of managing discontinuous in Euler angle-based pose estimation.

### 5.1 ABLATION STUDIES

Since Pavllo et al. (2019); Zheng et al. (2021); Zhao et al. (2023; 2019) stand for typically network architecture, we conducted comparisons for each method under the following settings:
**A**. Euler angles are predicted via the pose estimation method alone, without any auxiliary.
**B**. Euler angles are predicted via the results of an external classifier.
**C**. Euler angles and conditional information are predicted via the improved framework.
**D**. The improved framework predicts the values of sine and cosine of Euler angles along with the conditional information, and the final Euler angles are computed by the arctangent function.

Table presents the MPJASE losses of the 4 pose estimation methods below, each method are tested in our ablation settings from A to D. For Pavllo et al. (2019), our approach consistently reduces the error across all actions, leading to an average improvement of **0.60°**, with notable gains in *Walk*

Table 1: Quantitative comparison of MPJASE under ablation settings from A to D on our Euler angle Dataset.

| Methods | Ablation | Walk | Sit | Run | Jump | Squat | Torso | Arm | Leg | Avg |
|---|---|---|---|---|---|---|---|---|---|---|
| Pavllo et al. (2019) | A | 5.73 | 8.00 | 5.99 | 6.50 | 5.46 | 6.47 | 6.00 | 7.26 | 6.43 |
| | B | 5.16 | 7.32 | 5.50 | 5.44 | 5.28 | 6.31 | 5.81 | 6.61 | 5.93 |
| | C | 5.25 | 7.87 | 5.87 | 5.84 | 5.41 | 6.45 | 5.85 | 6.56 | 6.14 |
| | D | **5.08** | **7.22** | **5.43** | **5.42** | **5.21** | **6.29** | **5.71** | **6.25** | **5.83** |
| Zheng et al. (2021) | A | 6.90 | 8.49 | 7.77 | 7.05 | 5.81 | 8.07 | 6.31 | 8.61 | 7.38 |
| | B | 5.15 | 7.41 | 5.32 | 5.18 | 5.60 | 6.58 | 6.13 | 7.10 | 6.06 |
| | C | 5.66 | 7.42 | 7.45 | 6.88 | 5.65 | 6.51 | 6.22 | 7.51 | 6.66 |
| | D | **4.42** | **7.32** | **5.21** | **4.98** | **5.19** | **5.71** | **5.66** | **6.85** | **5.67** |
| Zhao et al. (2023) | A | 9.19 | 11.44 | 9.72 | 8.23 | 6.77 | 9.01 | 7.52 | 9.65 | 8.94 |
| | B | 5.90 | 8.38 | 5.62 | 6.35 | 6.75 | 7.47 | 6.96 | 7.79 | 6.90 |
| | C | 6.82 | 9.22 | 6.35 | 7.51 | 6.90 | 9.21 | 6.81 | 8.12 | 7.62 |
| | D | **5.68** | **8.21** | **5.52** | **6.28** | **6.71** | **7.41** | **6.88** | **7.68** | **6.80** |
| Zhao et al. (2019) | A | 10.32 | 11.25 | 10.84 | 11.13 | 10.37 | 11.34 | 10.27 | 10.96 | 10.81 |
| | B | 8.59 | 10.77 | 8.69 | 9.20 | 9.20 | 9.90 | 8.58 | 9.78 | 9.34 |
| | C | 9.51 | 11.58 | 9.51 | 9.90 | 10.15 | 9.54 | 9.33 | 9.91 | 9.93 |
| | D | **8.42** | **10.41** | **7.84** | **7.13** | **9.17** | **8.71** | **8.24** | **8.96** | **8.61** |

(0.65°) and *Sit* (0.78°). Zheng et al. (2021) exhibits even stronger benefits, achieving an average reduction of **1.71°**, with pronounced improvements on dynamic actions such as *Walk* (2.48°) and *Run* (2.56°). Zhao et al. (2023) also demonstrates substantial improvements (average **2.14°**), particularly on *Walk* (3.51°) and *Run* (4.20°). Although Zhao et al. (2019) has the largest baseline error, it still gains a meaningful average reduction of **2.20°**, most clearly reflected in *Run* (3.00°) and *Jump* (4.00°).

Overall, actions involving larger periodic rotations—such as *Walk*, *Run*, and *Jump* show the most prominent improvements, with average reductions of **2.03°**, **2.03°**, and **2.00°**, respectively. In contrast, more static or locally constrained actions like *Squat* and *Torso* exhibit relatively smaller improvements of **0.61°** and **1.17°**. These results confirm that our method is particularly effective in handling dynamic rotational movements without introducing any degradation in performance.

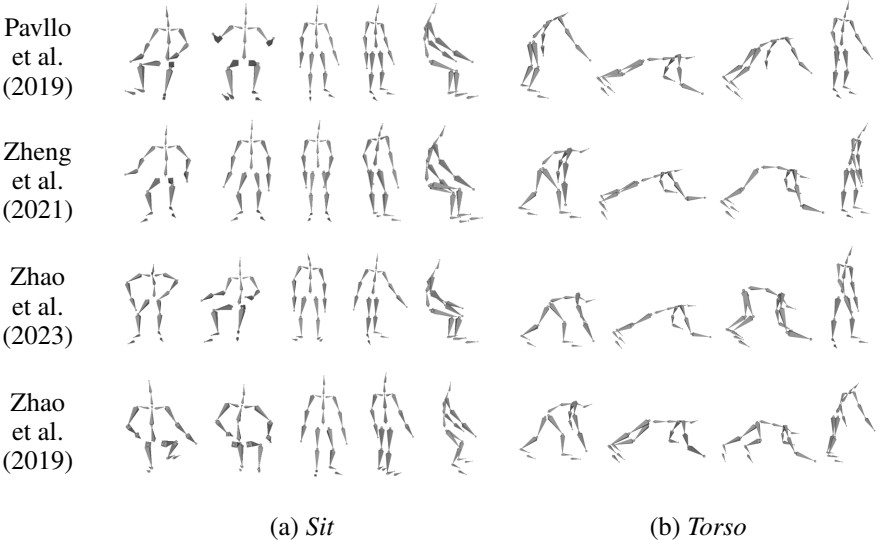

Pavllo et al. (2019)

Zheng et al. (2021)

Zhao et al. (2023)

Zhao et al. (2019)

(a) *Sit*          (b) *Torso*

Figure 7: Comparison on *Sit* and *Torso* actions under ablations set D.

We present visualizations of complex poses under the D setting, with actions selected from the *Sit* and *Torso* categories, specifically a seated posture and a push-up posture. As is shown in 7, the results demonstrate that predicting Euler angles via their sine and cosine values, followed by

recovery using the arctangent function, effectively resolves rotational discontinuities even in these challenging poses.

# 6 CONCLUSION

This paper presented a Euler angle-based method for 3D human pose estimation, addressing the discontinuity problem of wrap-around in body self-rotation angle prediction. Our method maps each Euler angle to its sine-cosine representation on the unit circle and effectively transform the rotation representation from the SO(3) manifold to the SO(6) manifold by leveraging the orientation cue as an additional prior. Experiments demonstrate significant performance gains across current CNN-, GCN-, and Transformer-based model. Our study demonstrates that explicitly modeling conditional information can effectively enhance the prediction continuity in rotation representation and the generalization of human pose estimation. Future work will extend the method to parametric human body models and explore novel neural network structures to improve the prediction accuracy, Overall, incorporating orientation priors enables Euler rotation angles to be learned reliably, effectively resolving the wrap-around problem.

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

## A  DATASET

In this study, we constructed a 3D human pose dataset represented by 3D Euler angle sequences. Several participants were enrolled and performed motions after training within a calibrated capture space of $5m{\times}3.2m{\times}2.5m$. High-precision body motion was captured using an OptiTrack system, with synchronized video sequences were recorded with an Orbbec Femto Bolt camera. Each subject was equipped with 41 reflective markers, the trajectories of which were subsequently converted into Euler angle rotations for 17 body joints.

The dataset covers a diverse set of actions, including *Walk*, *Sit*, *Run*, *Jump*, *Squat*, *Torso*, *Arm*, and *Leg*, spanning extreme rotational angles across all joints.

(1) **Euler angle dataset**: We use a self-collected dataset consisting of 117,325 training frames and 32,219 testing frames. The human body in this dataset consists of 17 joints, with each joint represented by a single Euler angle following the XYZ rotation order. During data acquisition, the optical axis of the camera was aligned with the $-z$ direction of the motion capture coordinate system, ensuring that horizontal body rotations are accurately projected onto the imaging plane. Examples of captured actions are illustrated in Figure 8, while the motion capture setup and equipment are shown in Figure 9. The number of frames per action is summarized in Table 2.

(2) **Orientation dataset**: The orientation scalar is a binary scalar (0 or 1), with one scalar value corresponding to each frame. The value is determined by the sign of the projected angle of the root joint's Euler angle on the 2D horizontal plane: positive angles are assigned 1, and negative angles 0. The training and testing splits for this orientation data, which are consistent with those of the Euler angle dataset, as summarized in the Table 2.

(3) **2D keypoint data**: The 2D keypoints for all video frames are obtained using inference from Detectron2.

Table 2: Frame distributions of the training and test sets.

|  | Walk | Sit | Run | Jump | Squat | Torso | Arm | Leg | Sports | Sum |
|---|---|---|---|---|---|---|---|---|---|---|
| **Training set** | 13993 | 17843 | 2168 | 5253 | 2613 | 11027 | 26031 | 12246 | 20151 | 117325 |
| **Test set** | 1229 | 1963 | 1040 | 988 | 275 | 959 | 2429 | 1112 | 22924 | 32219 |

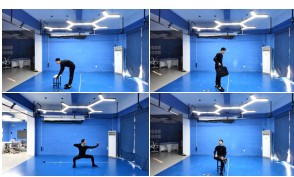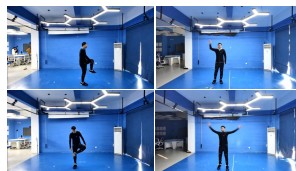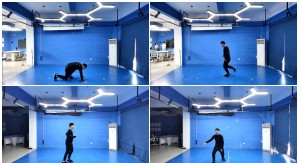

Figure 8: Examples of actions in our datase

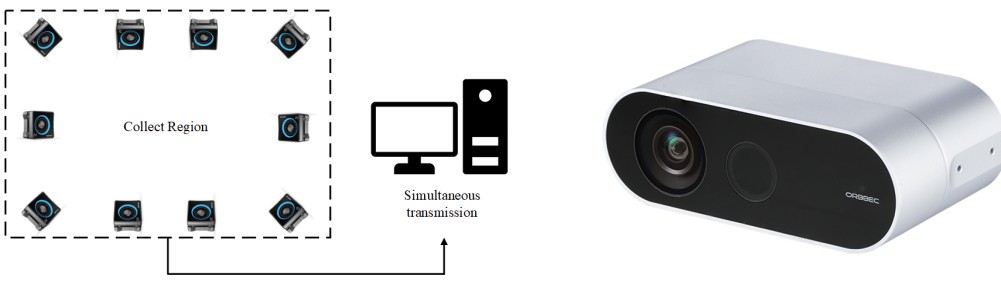

(a) OptiTrack Motion capture system    (b) Orbbec Femto Bolt Camera

Figure 9: The motion capture system and camera for our dataset collection

## B TRAINING DETAILS

To accommodate training with Euler angle rotation-based human pose dataset, we applied uniform preprocessing to all methods.: (1) Preprocessing operations originally designed for position learning were replaced with angle-based preprocessing to match the Euler angle representation. (2) The input channel dimensions of the networks were adjusted. The original networks were designed for 2D joint inputs; additional channels were added to incorporate the conditional Euler angle priors.

All baseline methods are implemented with their official default configurations or the authors' recommended setups, while our proposed module is integrated into each of them for evaluation. To ensure fairness, we follow the original training protocols as closely as possible and only make minimal adjustments where necessary. Specifically, Pavllo et al. (2019) is trained with five layers of dilated convolutions (kernel size 3) and a batch size of 1024, without strided convolutions, causal convolutions, or semi-supervised learning, for a total of 60 epochs. Zheng et al. (2021) is trained with $f = 27$ using its default settings until convergence, and Zhao et al. (2023) adopts $f = 27$ with similar hyperparameters and a comparable training schedule. Liu et al. (2020b) is trained for 40 epochs using its default configuration. Liu et al. (2020a) is trained with a batch size of 128 and default architectural parameters for a moderate number of epochs to ensure stable convergence.Zhao et al. (2019) is trained with the non-local module enabled for 90 epochs to capture spatiotemporal dependencies. For all methods, the initial learning rate is set to 0.001 and decayed by a factor of 0.95 per epoch, consistent with the respective default schedules.

Table 3: Training configurations for all 3D human pose estimation methods.

| Method | Core Settings | Batch Size | Epochs | Special Modules |
|---|---|---|---|---|
| Pavllo et al. (2019) | $arc = 3, 3, 3, 3, 3$ | 1024 | 60 | – |
| Zheng et al. (2021) | $f = 27$ | default | 70 | – |
| Zhao et al. (2023) | $f = 27$ | default | 70 | – |
| Liu et al. (2020b) | default | default | 40 | – |
| Liu et al. (2020a) | $arc = 3, 3, 3$ | 128 | 60 | – |
| Zhao et al. (2019) | default | default | 90 | non-local module |

*Note: All methods use initial learning rate lr = 0.001 with decay factor 0.95 per epoch.*

