# OpenReview forum: "Monocular 3D Human Pose Estimation via Euler Angles"
_ICLR.cc/2026/Conference — Submitted to ICLR 2026_

### Official Review · Reviewer_zrhc · 2025-10-24

**Soundness:** 1
**Presentation:** 3
**Contribution:** 1
**Rating:** 2
**Confidence:** 4

**Summary:**

The paper proposes a 3D human pose estimation method from sequences of 2D human pose, aiming to address the discontinuity problem in Euler angle–based joint rotation representations. The paper models the body’s self-rotation as a helical trajectory and project it onto a 2D plane, where the rotation space is divided into two discrete orientation intervals (e.g., facing left vs. right). A classifier predicts the orientation condition, which is then appended to the 2D keypoints as an additional input feature. This conditional signal is intended to help the network produce continuous Euler angle predictions across rotation boundaries and mitigate wrap-around discontinuities.

**Strengths:**

The paper is clearly written and easy to follow. The overall presentation is also nice. The proposed method appears effective in producing smoother and more accurate joint orientation predictions. Moreover, the conditional Euler angle module is designed as a plug-and-play component, making it compatible with a wide range of existing 3D pose estimation frameworks.

**Weaknesses:**

The paper has several key weaknesses:

-  **Weak motivation:** The paper does not clearly motivate the choice of Euler angles over more established rotation representations such as 6D (Zhou et al., 2019) or quaternions. The claim that alternative representations introduce excessive parameter complexity is unconvincing, as the overhead is minimal and these formulations are more robust and geometrically consistent with the SO(3) group of rotations. The discontinuity problem in Euler angles is well known, which is exactly the reason why such alternatives are widely adopted. Moreover, projecting angles onto a 2D plane and introducing a binary orientation condition does not enforce SO(3) manifold constraints, potentially resulting in mathematically invalid or non-physical rotations.

- **Issues with the loss formulation:** The use of an L1 loss on Euler angles fails to respect the underlying SO(3) geometry, leading to ambiguities and inconsistent rotational behavior. The paper does not clarify how the proposed method mitigates gimbal lock or ensures rotation invariance.

- **Insufficient dataset details:** The paper provides limited information about the proposed dataset—such as the number of subjects, recording conditions, or quality control measures. The authors also do not state whether they intend to make the dataset public.

- **Limited and unclear evaluation:** All experiments are conducted on the proposed dataset, with no evaluation on public benchmark datasets like Human3.6M. While these datasets may not provide Euler angles directly, they include rotation matrices from which Euler angles can be easily derived. This omission significantly weakens the empirical validation. Moreover, improvements are reported only in angular metrics (MPJASE) without relating them to common benchmarks such as Mean Per Joint Position Error (MPJPE) in millimeters, making it difficult to assess practical impact. Although the method is integrated into several existing frameworks (e.g., Zhao et al., 2019), all comparisons remain within the Euler-angle domain, with no results against alternative rotation representations like 6D or quaternions. It also remains unclear how the integration is implemented (e.g., whether prediction heads are modified), raising doubts about whether the observed gains primarily result from the conditional left/right orientation classification rather than a fundamentally better representation.

- **Lack of ablation and quantitative analysis:** The effect of the conditional classification is shown only qualitatively, with no quantitative ablation to isolate its contribution—an important omission.

**Questions:**

As mentioned in the weakness section, there are multiple major weaknesses as such I have the following questions:

- To what extent does the minor parameter overhead of other rotation representations (e.g., 6D or quaternions) justify the decision to predict Euler angles, which are known to suffer from discontinuities and gimbal lock?

- How does the proposed 2D projection and binary conditioning ensure that the predicted rotations remain valid on the SO(3) manifold rather than producing geometrically invalid or impractical rotations?

- How does the L1 loss on Euler angles handle wrap-around discontinuities and gimbal lock while ensuring that the predicted rotations remain consistent with the SO(3) manifold?

- Could you provide more details about the dataset (e.g., number of subjects and quality control)? Will it be released publicly?

- Why were no evaluations conducted on standard benchmarks (e.g., Human3.6M), given that Euler angles can be derived from rotation matrices?

- How do improvements in angular metrics (MPJASE) relate to positional metrics like MPJPE?

- When integrating into existing frameworks (e.g., Zhao et al., 2019), were prediction heads modified to output Euler angles?

- Have you compared the conditional Euler formulation directly with other rotation representations under identical settings?

- Can you provide quantitative ablations showing the contribution of the conditional classifier to the final performance?

---

> ### Author Response · Authors · 2025-12-03
>
> Dear Reviewer,
>
> Thank you very much for taking the time to review our submission and for providing valuable feedback. We greatly appreciate your thoughtful comments, which have been extremely helpful in improving our work.
>
> In response to the limitations you identified in the original two-stage framework, we have fundamentally revised our approach and redesigned the method as a one-stage framework. Consequently, the manuscript has undergone substantial restructuring and extensive modifications. Due to the extensive modifications, including additional figures and detailed explanations, the responses exceed the space available in the review box. Therefore, we have uploaded a supplementary document containing the full Rebuttal. Please refer to the corresponding supplementary material for filename of "reply to reviewer1.md" and "iclr2026_revised and highlight_manuscript.pdf" in Supplementary Material.zip fileDear Reviewer,
>
> Thank you very much for taking the time to review our submission and for providing valuable feedback. We greatly appreciate your thoughtful comments, which have been extremely helpful in improving our work.
>
> In response to the limitations you identified in the original two-stage framework, we have fundamentally revised our approach and redesigned the method as a one-stage framework. Consequently, the manuscript has undergone substantial restructuring and extensive modifications. Due to the extensive modifications, including additional figures and detailed explanations, the responses exceed the space available in the review box. Therefore, we have uploaded a supplementary document containing the full Rebuttal. Please refer to the corresponding supplementary material for filename of "reply to reviewer4.md" and "iclr2026_revised and highlight_manuscript.pdf" in Supplementary Material.zip file

---

### Official Review · Reviewer_7pF4 · 2025-10-26

**Soundness:** 2
**Presentation:** 3
**Contribution:** 2
**Rating:** 4
**Confidence:** 5

**Summary:**

The paper introduces a conditional Euler angle-based framework for monocular 3D human pose estimation that mitigates discontinuities in rotation prediction caused by Euler angle wrap-around. By projecting rotations into a 2D conditional space and conditioning on body orientation inferred from 2D keypoints, the method enables continuous self-rotation estimation. It can be integrated into existing CNN, GCN, and Transformer models, achieving smoother and more accurate pose predictions.

**Strengths:**

- By projecting the root joint's horizontal rotation onto a 2D plane and assigning a binary condition label, the work divides the rotation space into two half-circles: the right-facing interval $[0^\circ, 180^\circ]$ and the left-facing interval $[-180^\circ, 0^\circ]$. This helps the network learn separate continuity within each interval and mitigates the wrap-around discontinuity at $\pm180^\circ$.
- Given that the horizontal splitting trick proposed in the work should work in Euler angle space due to explicitly encoding the body's yaw orientation, and the fact that there is no existing dataset that records Euler angles, the authors collect a dataset of 150k frames to make training possible.
- The classifier study (Section 4.1) compares the failure cases of the Softmax classifier and the ResNet, and investigates the cases where both incorrectly classify the horizontal intervals.
- Evaluated on six separate baseline models, the work shows consistent improvement in MPJASE when the proposed method is applied to the baselines.

**Weaknesses:**

- The conditional splitting relies on a classifier to assign labels. If the classifier misclassifies a frame, as shown in Figure 6, then assuming the ground truth is 179°, the network might predict a value like -170°, and the proposed MPJASE will assign $|179 - (-170)| = 349^\circ$, while the person only rotated 11°. The L1 loss used in MPJASE may not be the optimal solution for periodic angular data.
- The work only reports results on the privately recorded dataset, and only MPJASE is presented. It is therefore unknown how the proposed method would affect other metrics, such as the Mean Per Joint Position Error (MPJPE), on public datasets.
- The division of the space into two intervals is coarse and might fail in complex poses, such as twisting the upper body.
- The work describes converting OptiTrack marker trajectories into Euler angles but does not specify how bone hierarchies were defined or which rotation conventions were used (e.g., ZYX, XYZ).
- Although Table 2 shows consistent improvement when the proposed method is applied, there is no experiment isolating the effect of the conditional classification alone or the augmented input representation, making it unclear which part of the improvement actually comes from the proposed method.

Minor issues:
- line 121: continues -> continuous
- line 204: fig 3(b) -> fig 3(a)
- line 269: MPJAE -> MPJASE
- unclear which GPU used in table 1
- It would be better to refer to Fig. 5(a) in the first paragraph of Section 3.2.2 when describing the concatenation.

**Questions:**

- Why is geodesic loss not used in MPJASE instead of L1 loss to address discontinuity issues?
- The classifier proposed in the paper uses 2D keypoints as input. How is ResNet18 used as a classifier if it expects image inputs? If 2D keypoints are represented as heatmaps, how is the inference speed still exceptionally high?

---

> ### Author Response · Authors · 2025-12-03
>
> Dear Reviewer,
>
> Thank you very much for taking the time to review our submission and for providing valuable feedback. We greatly appreciate your thoughtful comments, which have been extremely helpful in improving our work.
>
> In response to the limitations you identified in the original two-stage framework, we have fundamentally revised our approach and redesigned the method as a one-stage framework. Consequently, the manuscript has undergone substantial restructuring and extensive modifications. Due to the extensive modifications, including additional figures and detailed explanations, the responses exceed the space available in the review box. Therefore, we have uploaded a supplementary document containing the full Rebuttal. Please refer to the corresponding supplementary material for filename of "reply to reviewer3.md" and "iclr2026_revised and highlight_manuscript.pdf" in Supplementary Material.zip file

---

### Official Review · Reviewer_tVh2 · 2025-11-01

**Soundness:** 2
**Presentation:** 4
**Contribution:** 3
**Rating:** 4
**Confidence:** 5

**Summary:**

This paper focuses on solving the discontinuity problem in Euler angle–based human 3D joint rotation prediction from a single RGB image. They formalize Euler angle discontinuities as a wrap-around problem on a 3D helix, revealing that the discontinuity is a topological artifact rather than a numerical error. They project the continuous rotation space onto a 2D plane, discretize it into angular intervals, and use a conditional orientation classifier to provide body orientation priors for regression.
The conditional label is fused with 2D keypoints as an additional input channel, forming an extended keypoint representation. This allows the model to predict Euler angles continuously without modifying the network backbone, enabling a true plug-and-play design for existing monocular 3D human pose estimation frameworks (e.g., CNN-, GCN-, and Transformer-based models).
Experiments are conducted on a 3D human pose dataset they collected using an OptiTrack system. Human poses are represented by Euler angle rotations. Integrating the proposed method shows consistent improvement of 6 3D human pose estimation methods on their angle dataset.

**Strengths:**

1. The paper provides a theoretical explanation of the discontinuity phenomenon in Euler angle representation by modeling body rotation as a 3D helix, which intuitively reveals the geometric cause of the wrap-around issue. By combining high-dimensional projection with conditional discretization, the method achieves continuous Euler angle learning in a compact and learnable 2D space.
2. The design of a plug-and-play module allows for easy integration into existing 3D pose estimation networks without structural modifications, improving stability and continuity.
3. Experimental (both quantitative and qualitative) results confirm the method’s strong architecture-agnostic generalization and efficient transferability between position- and rotation-based formulations.

**Weaknesses:**

1. The experiments are conducted only on the authors’ self-collected dataset. While this helps validate the proposed framework under controlled settings, the lack of evaluation on public benchmarks such as Human3.6M or MPI-INF-3DHP limits the generalizability and persuasiveness of the results.
The authors mention that “existing public datasets do not provide Euler angle annotations,” which is true; however, Euler angles can be derived from the available rotation matrices or joint orientations provided by these datasets. Including such experiments, even through conversion, would greatly enhance the credibility and reproducibility of the method.
2. The potential impact of misclassification in the conditional orientation classifier and its influence on regression stability could be analyzed more thoroughly.
3. Lacking the comparisons with learning the continuous representation, like 6D representation.

**Questions:**

1.Explicitly defining 3D pose based on Euler angles at the beginning of the method makes it easier for readers to understand.
2.Further comparison with directly learning continuous angular representations can further highlight the value of this research.

---

> ### Author Response · Authors · 2025-12-03
>
> Dear Reviewer,
>
> Thank you very much for taking the time to review our submission and for providing valuable feedback. We greatly appreciate your thoughtful comments, which have been extremely helpful in improving our work.
>
> In response to the limitations you identified in the original two-stage framework, we have fundamentally revised our approach and redesigned the method as a one-stage framework. Consequently, the manuscript has undergone substantial restructuring and extensive modifications. Due to the extensive modifications, including additional figures and detailed explanations, the responses exceed the space available in the review box. Therefore, we have uploaded a supplementary document containing the full Rebuttal. Please refer to the corresponding supplementary material for filename of "reply to reviewer2.md" and "iclr2026_revised and highlight_manuscript.pdf" in Supplementary Material.zip file

---

### Official Review · Reviewer_CkVc · 2025-11-03

**Soundness:** 1
**Presentation:** 2
**Contribution:** 1
**Rating:** 2
**Confidence:** 5

**Summary:**

This paper proposes a method to handle the facing angle prediction errors that may happen in 3D human pose estimation from single view images. The proposed method is straightforward. It first estimates a scalar which is a rough estimation of the person's facing direction in the left or right half of the angle space. Then, this scalar is concatenated with the 2D pose key point estimations to give the final 3D pose estimation result. The authors show results that improve the angle continuity of the proposed method.

**Strengths:**

+ The proposed method is simple and can be combined with different existing methods.

**Weaknesses:**

1. The proposed method is highly dependent on the correctness of the orientation scalar estimated in the first step. If the rough pose estimation is wrong, the result is likely to be completely wrong.

2. The authors did not give a clear description about the orientation scalar network and how to guarantee its estimation quality.

3. It is not clear what softmax classifier means.

4. It is not clear what dataset is the classifiers such as Resnet18 in table 1 is trained on and tested on.

5. Since the experimental settings are not clearly defined, the experimental results are not convincing.

**Questions:**

Please address the concerns and questions in the weakness session.

---

> ### Author Response · Authors · 2025-12-03
>
> Dear Reviewer,
>
> Thank you very much for taking the time to review our submission and for providing valuable feedback. We greatly appreciate your thoughtful comments, which have been extremely helpful in improving our work.
>
> In response to the limitations you identified in the original two-stage framework, we have fundamentally revised our approach and redesigned the method as a one-stage framework. Consequently, the manuscript has undergone substantial restructuring and extensive modifications. Due to the extensive modifications, including additional figures and detailed explanations, the responses exceed the space available in the review box. Therefore, we have uploaded a supplementary document containing the full Rebuttal. Please refer to the corresponding supplementary material for filename of "reply to reviewer1.md" and "iclr2026_revised and highlight_manuscript.pdf" in Supplementary Material.zip file

---

### Meta-Review · Area_Chair_vGza · 2026-01-09

**Summary:**

This paper proposes a 3D human pose estimation method from sequences of 2D human pose, aiming to address the discontinuity problem in Euler angle–based joint rotation representations.  The method first estimates a scalar which is a rough estimation of the person's facing direction in the left or right half of the angle space. Then, this scalar is concatenated with the 2D pose key point estimations to give the final 3D pose estimation. The design of the conditional Euler angle module as a plug-and-play component is nice because it allows easy integration into existing 3D pose estimation networks without structural modifications.  However, lots of concerns were raised by the reviewers.  They include impact of inaccurate direction-scalar estimation on 3D pose estimation, questionable loss formalization in the proposed method, unclear details of experiments, and insufficient evaluation.  The rebuttal provided by the authors is not standard in the sense that the authors did not directly resolve the concerns on the proposed method; instead, revised the proposed method to a new one and argued the concerns based on the new method.  In the rebuttal, the authors fully agreed to the drawback pointed out by the reviewers and then replaced the proposed two-stage framework with a unified one, which is quite unusual.  The objective of the rebuttal is to resolve the unclear/ambiguous issues related to the originally proposed method but not to develop a new method.  If a new method is developed during the rebuttal phase, the authors should propose it as a new submission: the current change is far beyond the revision through the rebuttal.  For this, this paper should be rejected.

**Reviewer Concerns:**

The impact of the direction-scalar estimation accuracy on the pose estimation is a crucial concern raised by all the reviewers, which was not resolved directly. The reviewers suggested evaluation on public benchmarks such as Huam3.6M or MPI-INF-3DHP but not only on the authors’ self-collected dataset, which was not properly addressed.  At least evaluation on Huma3.6M should have been addressed.  The concern of the usage of an L1 loss on Euler angles was raised because it may fail to respect the underlying SO(3) geometry, which was not resolved directly; instead, the authors proposed a new framework.  Details about the experimental setting and dataset were addressed.

**Reviewer Scores:**

All the reviewers would keep their initial scores or change to lower scores.  This is because, as written above, the rebuttal has not directly addressed the concerns based on the originally proposed method.

---

### Decision · Program_Chairs · 2026-01-26

Reject